# Learning Multimodal Behaviors from Scratch with Diffusion Policy Gradient

**Zechu Li**[1,2]    **Rickmer Krohn**[1,3]    **Tao Chen**[2]    **Anurag Ajay**[2]
**Pulkit Agrawal**[2]    **Georgia Chalvatzaki**[1,3]
[1]Technical University of Darmstadt    [2]Massachusetts Institute of Technology    [3]Hessian.AI
{zechu.li,rickmer.krohn}@stud.tu-darmstadt.de
{taochen,aajay,pulkitag}@mit.edu
{georgia.chalvatzaki}@tu-darmstadt.de

## Abstract

Deep reinforcement learning (RL) algorithms typically parameterize the policy as a deep network that outputs either a deterministic action or a stochastic one modeled as a Gaussian distribution, hence restricting learning to a single behavioral mode. Meanwhile, diffusion models emerged as a powerful framework for multimodal learning. However, the use of diffusion policies in online RL is hindered by the intractability of policy likelihood approximation, as well as the greedy objective of RL methods that can easily skew the policy to a single mode. This paper presents Deep Diffusion Policy Gradient (DDiffPG), a novel actor-critic algorithm that learns *from scratch* multimodal policies parameterized as diffusion models while discovering and maintaining versatile behaviors. DDiffPG explores and discovers multiple modes through off-the-shelf unsupervised clustering combined with novelty-based intrinsic motivation. DDiffPG forms a multimodal training batch and utilizes mode-specific Q-learning to mitigate the inherent greediness of the RL objective, ensuring the improvement of the diffusion policy across all modes. Our approach further allows the policy to be conditioned on mode-specific embeddings to explicitly control the learned modes. Empirical studies validate DDiffPG's capability to master multimodal behaviors in complex, high-dimensional continuous control tasks with sparse rewards, also showcasing proof-of-concept dynamic online replanning when navigating mazes with unseen obstacles. Our project page is available at `https://supersglzc.github.io/projects/ddiffpg/`.

## 1 Introduction

Reinforcement learning (RL) for continuous control has experienced significant advancements during the last decade, reshaping its applicability to domains like game playing [65, 27, 4], robotics [26, 33], and autonomous driving [72, 12]. However, most RL algorithms choose to parameterize policies as deep neural networks with deterministic outputs [47, 21] or Gaussian distributions [62, 25], limiting learning to a single behavior mode. Moreover, the standard exploration-exploitation schemes can easily make a policy greedy towards one mode, in which the algorithm keeps exploiting to maximize its objective. The aforementioned issues hinder the possibility of training agents that successfully solve a task while showcasing versatility of behaviors—a property intuitive to intelligent systems like humans, who can exhibit resourcefulness for completing a task, even amidst unprecedented events.

Learning a multimodal policy has several practical applications. First, such a policy that learns many solutions to a task is useful when acting in non-stationary environments. Imagine that a routine path from the office to home is unexpectedly blocked; one needs to choose an alternate route. A policy that encompasses multiple solutions can better navigate such changing conditions, offering flexibility

38th Conference on Neural Information Processing Systems (NeurIPS 2024).

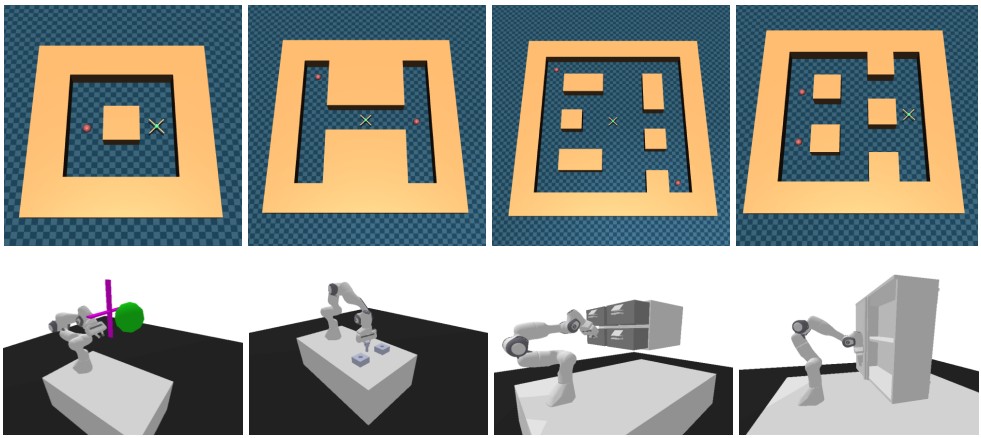

Figure 1: We design (Top) four AntMaze tasks; *AntMaze-v1*, *AntMaze-v2*, *AntMaze-v3*, *AntMaze-v4*, and (Below) four robotic tasks *Reach*, *Peg-in-hole*, *Drawer-close*, and *Cabinet-open* that have a high degree of multimodality.

for replanning or serving as prior for hierarchical RL. Second, while searching for diverse solutions, a multimodal policy continues exploring even after finding a viable solution, thereby facilitating agents to escape local minima. Additionally, multimodal policies hold great promise for continual learning scenarios; they can effectively parameterize complex action distributions when new skills or solutions are introduced, potentially mitigating the issue of catastrophic forgetting [59].

Recently, Diffusion Models [30, 68], a novel class of generative models acclaimed for impressive image generation results [38, 60], have been proposed as a powerful parameterization for policy learning. They have been extensively applied in the areas of learning from demonstrations [15, 58], offline RL [73, 71], and learning for trajectory optimization [34, 46]. One advantage of training diffusion policies *offline* is its ability to model multimodal datasets, which usually comprise trajectories stemming from suboptimal policies or various human demonstrations. However, only a few studies explored these models for *online* RL, which focus on the formulation of the training objectives for policy optimization via diffusion and showcase improved sample efficiency [55, 74, 17]. Notably, none of them studied the inherent multimodality within diffusion policies, nor have they explicitly tackled the challenge of exploration for discovering and learning multiple behavioral modes online.

In this paper, we introduce *Deep Diffusion Policy Gradient* (DDiffPG), a novel actor-critic algorithm for training multimodal policies parameterized as diffusion models from scratch. Specifically, we aim to learn a policy that is capable of employing various strategies to accomplish a task. Unlike existing multimodal methods that condition on latent variables, DDiffPG emphasizes the explicit discovery, preservation, and improvement of behavioral modes, and for that, we decouple exploration and exploitation. For exploration, we apply novelty-based intrinsic motivation and utilize an unsupervised hierarchical clustering approach to discover modes, being a priori agnostic to the possible number of modes. For exploitation, we introduce mode-specific Q-functions to ensure independent improvements across modes and construct multimodal data batches to preserve the multimodal action distribution. We empirically evaluate our method on high-dimensional and continuous control tasks with sparse rewards, comparing to SoTA baselines, verifying DDiffPG's capbility to master multimodal behaviors. Additionally, we demonstrate the usability of our multimodal policies in an online replanning application in non-stationary mazes with unseen obstacles.

Our paper makes **four contributions**. First, we introduce **diffusion policy gradient**, a novel way to train diffusion models following the RL objective. Second, we present DDiffPG, a **new actor-critic algorithm for training diffusion policies from scratch** while discovering and preserving multimodal behaviors. Third, we achieve **explicit mode control** by policy conditioning on a mode-specific latent embedding during training, shown to be beneficial for online replanning. Finally, we design a series of **new challenging robot navigation and manipulation tasks** with a high degree of multimodality, serving as a testbed for multimodal policy learning, as shown in Fig. 1.

## 2 Related Work

Policy learning via reinforcement learning for continuous control has exploded thanks to improved algorithms learning complex skills with *online* RL [25, 5], while significant effort has been made to provide improved methods for *offline* RL to take advantage of demonstrations and fixed datasets [40, 44]. However, the policy parameterization in both settings considers policies that can only model a single behavioral mode, e.g., by having a deterministic output [66, 5] or by learning a Gaussian distribution. Recently, the use of transformer models enabled learning for control through offline data as sequence modelling [14, 57, 31], which, through language conditioning [7, 23], can handle multi-goal behavior generation; though, it is unclear if these models can explicitly exhibit behavior diversity per goal. Similarly, goal-conditioned RL methods condition policies to various goals, but each goal-conditioned policy suffers from the single-mode modeling, prevalent in typical deep RL algorithms [2, 52, 10, 53].

**Diffusion policy as universal skill representation**  Diffusion models [30] have recently emerged as a promising parameterization for robust and multimodal robot learning. Early works used diffusion models for trajectory optimization [34, 69, 46], and as a policy for behavioral cloning [15]. Diffusion policy has soon shown its power as a universal skill representation [76] that allows learning complex landscapes of multimodal behaviors [11, 28], goal-conditioned [58, 61] or language-conditioned ones [24, 13]. Further, the expressivity of diffusion models has proven beneficial for offline RL [73, 71]. Nevertheless, the approaches for online RL with diffusion policy optimization are scarce [55, 74, 17], while these works do not study multimodality preservation within diffusion models, nor have they explicitly tackled the challenge of learning multimodal policies online. In this work, we present a new framework that takes advantage of diffusion policy as a universal skill representation model and introduces an algorithm that allows multimodal policy learning from scratch.

**Unsupervised skill discovery**  Multimodal behavior learning has been addressed through skill discovery approaches [9, 42, 32, 54, 36], either from offline data [64] or through unsupervised RL [19, 41], which usually exploit variational inference for mode/skill discovery [43] with policy conditioning on latent vectors, or by forming rewards or goals to be achieved by different low-level policies [39, 49]. Hierarchical learning methods, e.g., relying on options learning [3, 1], usually depend on state-specific mode discovery that conditions a low-level policy or triggers different skills [45, 35]; but the policy usually reaches each goal greedily without exhibiting versatile behaviors.

## 3 Diffusion Policy Gradient

It is not straightforward to apply training approaches of popular deep RL methods to train a diffusion policy online. First, it is practically intractable to approximate the policy likelihood [37]. Second, directly backprogating Q-values to the diffusion policy, like Q-learning methods [25, 47], is not viable—the Markov chain may lead to vanishing gradients [55].

To solve these issues, we first present diffusion policy gradient, a novel approach to training diffusion policies with the RL objective that is also suitable for learning multimodal behaviors. Similarly to DPG [66], we compute $\nabla_a Q(s, a)$ given a state $s$ and action $a$. However, we choose not to directly use the action gradient to optimize the policy, as this leads to vanishing gradients and instability. We rather obtain a target action $a^{target}$ via $a^{target} \leftarrow a + \eta \nabla_a Q(s, a)$, $\eta$ being a suitable learning rate. We can, then, train the diffusion policy based on the transitions in the datasets $\mathcal{D}$ using a behavioral cloning (BC) objective

$$\mathcal{L}(\theta) = \mathop{\mathbb{E}}_{t \sim [1,T], (a_0^{target}, s) \sim \mathcal{D}, \epsilon \sim \mathcal{N}(\mathbf{0}, \mathbf{I})} \|\epsilon - \epsilon_\theta(a_t^{target}, s, t)\|, \tag{1}$$

where $t \sim [1, T]$ refer to the diffusion step, and $\epsilon$ is the diffusion noise — see Appx. A for a brief introduction to Diffusion Models.

**Implementation**  For every data collection, we store the transition $(s, a, a^{target}, r, s')$ in the buffer $\mathcal{D}$, where $a^{target}$ is initialized as $a$. For every policy update, we sample the state-action pair $(s, a^{target})$, obtain a new $a^{target}$ via gradient ascent, update the policy with equation 1 and replace the new $a^{target}$ in the buffer. Thus, our policy update does not change the training objective of the diffusion model and avoids the vanishing-gradient problem. Note that the action $a^{target}$ that is used to update the policy is not inferred from the current policy, which provides an opportunity to control the behaviors of the policy in an off-policy fashion and lay the foundation for learning multimodal behaviors later.

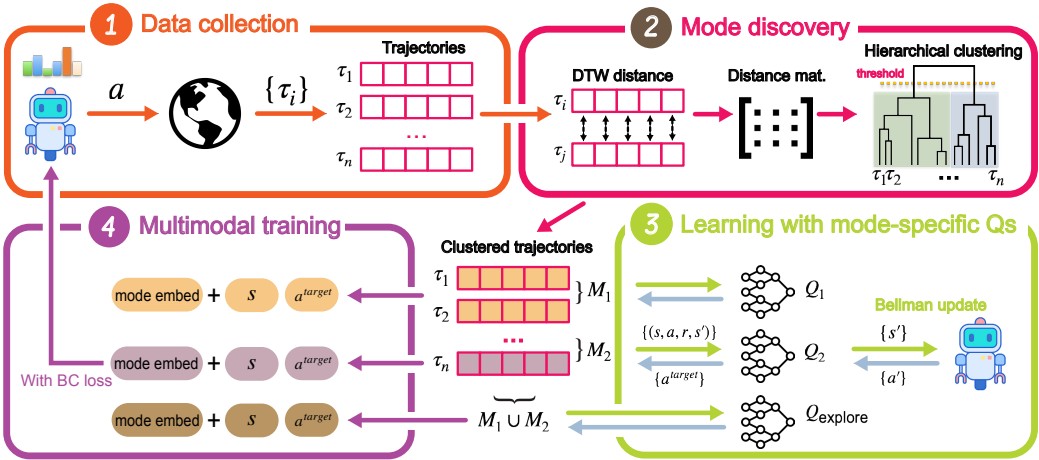

Figure 2: Overview of DDiffPG: (1) the agent interacts with the environment and collects a set of trajectories $\{\tau_i\}$. (2) Given a set of goal-reached trajectories, a DTW distance matrix is computed and used for hierarchical clustering to discover modes. (3) Each mode is associated with a set of trajectories, which is used exclusively to train mode-specific Q-functions and an exploration-specific $Q_{\texttt{explore}}$. (4) A multimodal batch is constructed by concatenating $(s, a^{target})$ pairs sampled from every mode and used for the diffusion policy update.

**Interpretation** The conventional diffusion model is trained using a dataset with supervised labels [30]. In *offline* decision-making, a diffusion policy predicts actions given states with the dataset providing both the state and the corresponding ground-truth action as the target (label). In contrast, our *online* setting does not provide a predefined ground-truth action but requires the discovery of good actions. To adapt without altering the supervised training framework of conventional diffusion models, we generate $a^{target}$ and consider it as the target. The action $a^{target}$, derived through gradient ascent, represents an improved choice based on the current Q-function. During training, we additionally store $a^{target}$ in the replay buffer and continuously update it based on its preceding values, ensuring a continuity of learning. Intuitively, we are chasing the target by replacing it with the newly computed $a^{target}$ in this *online* RL setting rather than learning a static target as in the *offline* setting.

**Relation to related works** Our formulation is most closely related to DIPO [74] but has two key differences. First, while DIPO also performs gradient ascent on the action, it replaces $a^{target}$ with the original $a$ from the buffer rather than retaining an additional $a^{target}$. Therefore, the transition no longer aligns with the current MDP dynamics and reward function due to the replacement of the original $a$. Given that DIPO is an off-policy algorithm, the reuse of these replaced transitions for training the Q-function could be problematic, as the agent is training values of actions that have not been actually played out in the environment and their true outcome (reward and next state) are unknown. Second, DIPO uses different batches for Q-learning and policy updates, which does not guarantee coherent updates. *Q-score matching* (QSM) [55] also computes $\nabla_a Q(s, a)$ and matches vector fields of $\nabla_a \log \pi(a|s)$ and $\nabla_a Q(s, a)$. Therefore, QSM is optimizing on the score level while ours focuses on the action level.

## 4 Learning Multimodal Behaviors from Scratch

We consider the problem of learning multimodal behaviors with online continuous RL, i.e., in the absence of initial demonstrations. In this section, we introduce *Deep Diffusion Policy Gradient* (DDiffPG), which builds off of three main ideas. First, we want to explicitly discover behavior modes and master them. Different modes should act differently at certain states and then diverge, therefore being distinguishable from trajectories/sequences of states. Second, we must prevent mode collapse once modes have been discovered, a common issue where the RL policy favors the mode with higher Q-values due to its inherent greediness. Finally, we expect the diffusion policy to capture and control the multimodal actions during evaluation. Fig. 2 provides an overview of the proposed method, and the pseudocode is available in Alg. 1.

## 4.1 Unsupervised Mode Discovery

**Novelty-based Exploration** In multimodal learning, it is necessary to explore diverse behaviors (i.e., modes). Effective exploration is important especially when considering challenging high-dimensional continuous control tasks with sparse rewards (cf. Fig. 1). We adopt a simple yet effective approach, namely using the difference between states' novelty as intrinsic motivation [75] to prompt exploration beyond already visited state-space regions. Following [8], we define the following intrinsic reward $r^{\text{intr}}(s, a, s') = \max(\text{novelty}(s') - \alpha \cdot \text{novelty}(s), 0)$, where novelty is parameterized as a Random Network Distillation (RND) [8] and $\alpha$ is a scaling factor. Note that while this intrinsic reward has been effectively verified for discrete state spaces, here we extend this exploration method to continuous domains, demonstrating its effectiveness for training diffusion policies.

**Hierarchical Trajectory Clustering** Our method explicitly identifies modes and masters them, unlike existing latent-conditioned approaches [32]. Given a collection of goal-reached trajectories, each consisting of a sequence of state-action pairs, we categorize them into clusters and consider each a behavior mode. In practice, we use an unsupervised hierarchical clustering approach [51]: as shown in Fig. 2(2), in the beginning, each trajectory is considered as a single cluster; then clusters within a small distance are progressively merged, continuing until a singular, unified cluster is formed.

Differently from clustering approaches like K-means [48] that require a predefined number of clusters (modes), we determine clusters using a distance threshold. Fortunately, given that the distance between different modes is usually large, hierarchical clustering is not as hyperparameter-sensitive as K-means. We show the clustering performance in Fig. 3 and the robust clustering threshold in Tab. 2. For distance metric, we utilize Dynamic Time Warping (DTW) [50] with task-prior information, e.g., robot positions [32]. An advantage is its applicability to variant-length trajectories, which removes the burden of padding or stitching trajectories. Note that our

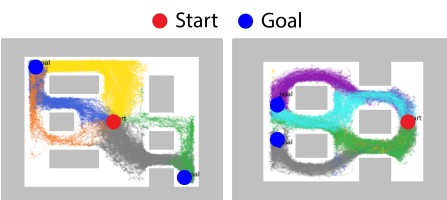

Figure 3: Hierarchical clustering on *AntMaze-v3* (Left) and *AntMaze-v4* (Right). Each color represents a mode.

method is agnostic to the particular clustering approach used, and it can be adapted to different (learning) approaches [63, 16]. For example, in Fig. 9(a), we provide an alternative clustering approach by training a VQ-VAE [70] to learn trajectory representations and using the codevectors for clustering, which learns meaningful representations suitable for our approach.

## 4.2 Mode Learning with Mode-specific Q-functions

To master multi-modes and improve them all together, we train a different Q-function per mode and construct multimodal data batches for policy learning.

**Learning mode-specific Q-functions** In RL, the objective to maximize expected return can skew the policy, leading to single-mode collapse. Let us consider the *AntMaze-v1* in Fig. 1; there are two viable paths to reach the goal. If the goal position is reached via the top path, the success bonus will propagate through the TD updates, meaning that the Q-values for the top path will increase and guide the policy to the top. Even though a simple diffusion RL [74, 73] method might initially explore both sides altogether, it will eventually end with a unimodal behavior determined by which side was explored first (cf. Fig. 11b & Fig. 12b). To address this issue, we propose training a mode-specific Q-function per discovered mode, allowing for parallel policy improvements across all modes. As shown in Fig. 2(3), trajectories that, for example, are categorized into two modes $M_1$ and $M_2$, will have two Q-functions $Q_1$ and $Q_2$, respectively. One needs to notice that such mode-specific Q-functions may capture suboptimal modes, which achieve the goal but require more steps hence yielding lower discounted returns. However, these suboptimal solutions may prove valuable in practical scenarios where the optimal action is infeasible, e.g., the routine path problem discussed in Sec. 1 and Sec. 5.5.

We additionally train a Q-function dedicated to exploration, which can be considered as an exploratory mode. This Q-function is trained exclusively with the intrinsic rewards and transitions from all trajectories, e.g., $M_1 \cup M_2$ in Fig. 2(3), regardless of which behavioral mode they represent. Such decoupling of exploration-exploitation on the Q-function level ensures that we continue exploring even after certain modes are well-learned, since our intrinsic reward only considers state novelty.

**Constructing a multimodal batch** The diffusion model's multimodality stems from the underlying multimodal distribution of the data. An intuitive strategy to obtain multimodality is to construct a multimodal training batch and feed it to the policy — each batch contains data from different modes. While the mode clustering is on the goal-reached trajectory level, it is essential to include data from unsuccessful trajectories as well. Practically, this is achieved by computing the distance between an unsuccessful trajectory and $N$ goal-reached trajectories randomly sampled from each cluster. The cluster with the smallest average distance is then designated as the final cluster for the unsuccessful trajectory (lines 8-14 in Alg. 2).

**Re-clustering** We perform re-clustering over trajectories at every $F$ iterations (see Tab. E). During this process, for each newly formed cluster, we assess its overlap with clusters identified in the previous clustering iteration. The new cluster then inherits Q-functions and $a^{target}$ from the preceding cluster that has the most overlap with. This ensures a continuity of learning and adaptation across successive re-clustering stages. The pseudocode is in Alg. 2, and implementation details are in Appx.C.

### 4.3 Mode Control via Latent Embeddings

Controlling the learned multimodal policy to exhibit specific behaviors rather than random generation is beneficial, especially in non-stationary test environments where certain modes may become nonviable. To achieve this, we propose conditioning the diffusion policy on mode-specific embeddings during training. As shown in Fig. 2(4), our method generates a unique latent embedding for each mode, which is then incorporated into the state information. Throughout the training process, we selectively include or mask (zero-out) these embeddings with a probability of $p$. By providing specific embeddings, we can, therefore, explicitly control the execution of desired modes or, alternatively, mask them to enable random mode selection. This technique affords several benefits:

- In non-stationary environments, planning approaches can empower the agent to navigate around undesirable modes through controlled mode selection, improving their success rate. We demonstrate an application for online replanning in Section 5.5.
- Our method learns multiple modes, some of which may be suboptimal, e.g., longer paths in the navigation problem. Given our knowledge of each mode's trajectories, we can estimate the expected return of each mode and select the one with the highest return to optimize performance.
- Since the exploratory mode has its unique embedding, we can control the exploration-exploitation tradeoff during training by adjusting the proportion of exploratory mode used in action generation at the data collection phase. Note that we exclude the exploratory mode to eliminate noisy exploratory behaviors at test time, leading to an increased success rate.

## 5 Experiments

In this section, we present a comprehensive evaluation of our method against SoTA baselines. First, we verify that DDiffPG can learn multimodal behaviors and discuss the performance compared to baselines. We then highlight the advantages of learning a multimodal policy in encouraging exploration and overcoming local minima. Finally, we provide ablations on important hyperparameters and showcase a practical application of replanning with such a multimodal policy. We run each experiment with five random seeds and plot their mean and standard error.

### 5.1 Setup

**Tasks** We evaluate our method on four AntMaze tasks [20] and four robotic control tasks [22], as shown in Fig. 1. Note that all tasks (1) are high-dimensional and continuous control tasks, e.g., in all *AntMaze* versions, the objective is to control the leg joints of an *ant* to reach the goal position; (2) contain multiple possible solutions, either with multiple goals or multiple ways that solve the task, e.g., in the *Reach* task, the robot arm can bypass the obstacle from four different directions; (3) are trained with sparse rewards, alleviating the need for engineering and reward shaping. The environment description is in Appx. D.

**Baselines** We consider the following baselines: (1) **DIPO** [74], which we have adapted to include the additional target action in replay buffer to ensure consistency in MDP dynamics and the reward function, as detailed in Section 3; (2) **Diffusion-QL** [73] and (3) **Consistency-AC** [17], which use diffusion model and consistency model for policy parameterization; (4) **Reparameterized Policy**

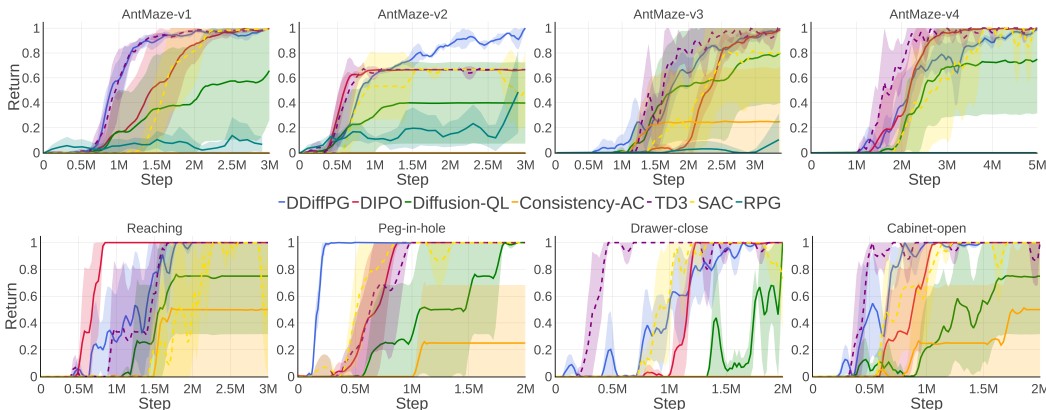

Figure 4: Performance of DDiffPG and baseline methods in the four AntMaze and robotic manipulation environments.

**Gradient (RPG)** [32], which uses a model-based approach with multimodal policy parameterization; (5) **TD3** [21]; (6) **SAC** [25]. For fair comparisons, we use double Q-learning [29] and distributional RL [6] for all baselines. Additionally, all baselines except RPG use the same intrinsic rewards as ours, while RPG has a similar built-in RND-based intrinsic reward as introduced in their paper. Hyperparameters are available in Tab. E.

## 5.2 DDiffPG Masters Multimodal Behaviors

We investigate whether DDiffPG can learn multimodal behaviors from scratch. We first perform observational evaluations and count the number of different modes over 20 episodes. As shown in Tab. 3 and Tab. 4, DDiffPG demonstrates consistent exploration and acquisition of multiple behaviors. For instance, in *AntMaze-v3*, multiple paths exist within the maze, but not all are the same length; DDiffPG is capable of learning and freely executing all these paths, including the suboptimal one — note that we call suboptimal a path with lesser discounted return than the shortest optimal one. However, assuming a goal-reaching success indicator all paths are successful. Nonetheless, the suboptimal issue can be mitigated given our ability to control the agent's behavior through the mode embeddings, as we discuss in Section 5.5. In *Cabinet-open*, the agent can move the arm to either layer and subsequently pull the door open. This contrasts with other methods, which fail to exhibit multimodal behavior. We observe that even policies parameterized as diffusion-based models, namely DIPO, Diffusion-QL, and Consistency-AC, can quickly collapse to a single mode and thereafter follow the greedy solution. This verifies the significance of our proposed method in capturing multimodality.

In Fig. 4, we see that DDiffPG has comparable performance to the baselines on all eight tasks while acquiring multimodal behaviors. In the AntMaze tasks, the sample efficiency of DDiffPG, DIPO, TD3 and SAC are similar: in *AntMaze-v1* and *AntMaze-v3*, TD3 and DDiffPG surpass the performance of others; in *AntMaze-v2* and *AntMaze-v4*, TD3 and DIPO are the most sample-efficient. In the manipulation tasks, a similar pattern emerges: DDiffPG leads *Peg-in-hole*, DIPO excels in *Reach*, and TD3 leads both *Drawer-close* and *Cabinet-open*. DDiffPG generally demonstrates lower sample efficiency than baselines in tasks that pose significant exploration challenges — this is expected since our method strives to discover multiple solutions. For example, in *AntMaze-v2*, the route to the top-left goal is more extended, and in *Reach*, the robotic dynamics make it difficult to explore the bottom paths. For simple exploration tasks, DDiffPG can achieve similar or even superior performance, as DDiffPG simultaneously explores the environment from multiple directions and the design of mode-specific Q-function effectively narrows the scope, facilitating faster convergence.

Diffusion-QL and Consistency-AC tend to lag behind as shown in Fig. 4. Both methods optimize the diffusion policy by backpropagating directly through the diffusion model, and we observed that their actor gradient may remain zero throughout the training in some seeds, resulting in a high variance (shadow area). In contrast, our diffusion policy gradient approach, which turns the training objective into minimizing the MSE loss w.r.t. the action target, demonstrates significantly greater stability. For

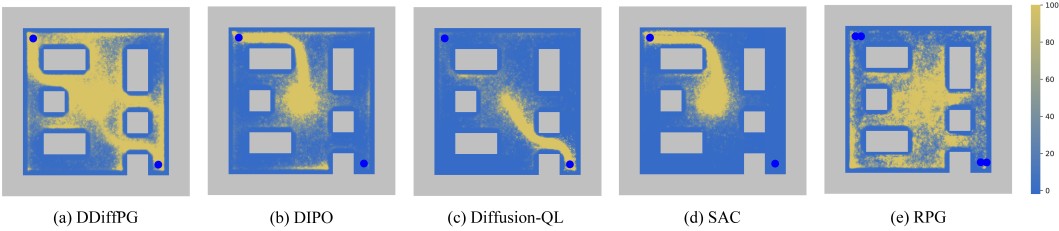

(a) DDiffPG     (b) DIPO     (c) Diffusion-QL     (d) SAC     (e) RPG

Figure 5: Exploration maps of DDiffPG and baselines in *AntMaze-v3*.

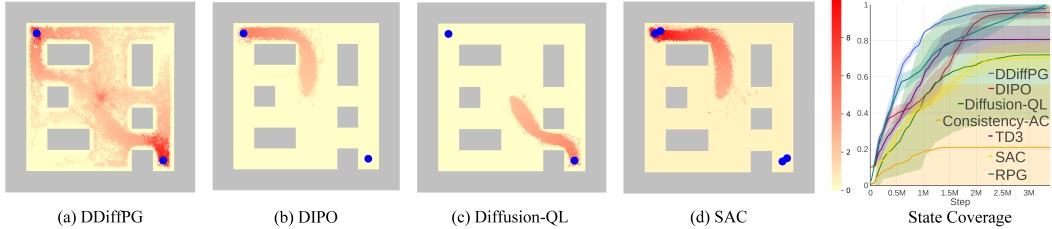

(a) DDiffPG     (b) DIPO     (c) Diffusion-QL     (d) SAC     State Coverage

Figure 6: (a)-(d) Q-value maps of baselines and DDiffPG and (Right) the state coverage in *AntMaze-v3*.

RPG, its performance illustrates that policy learning remains challenging despite demonstrating good exploration. One potential reason is that VAEs condition the policy on a latent variable, which offers a pathway to multimodality but sometimes leads to non-existing modes. This consideration led us to adopt a more straightforward yet effective clustering approach for explicit mode discovery.

### 5.3 Seeking of Multimodality Encourages Exploration and Overcomes Local Minima

**Observation 1.** *DDiffPG encourages exploration.*

We demonstrate the potential of DDiffPG in exploration using the exploration density maps and state coverage rates in the AntMaze environments. We discretize the maze and track the cell visitation. To avoid the dominance of high-density areas such as the starting positions, we set a max density threshold of 100, which means the cell has been visited at least 100 times. As shown in Fig. 5, in selected results for *AntMaze-v3*, DDiffPG explores multiple paths to the two separate goal positions, contrasting sharply with baselines that typically discover only a single path. For state coverage in Fig. 6, we measure the binary coverage of each cell and find that DDiffPG achieves a much higher coverage rate than baselines except RPG, **verifying that the continuous exploration capability of DDiffPG helps exploration**. While RPG achieves good exploration, it fails to solve the task as shown in Fig. 4. The maps for all AntMaze tasks and baselines are available in Appx. F.

**Observation 2.** *DDiffPG can overcome local minima.*

We showcase that DDiffPG effectively overcomes local minima when learning a multimodal policy. The key intuition is that unlike other methods that explore the first solution and collapse into it, DDiffPG continuously explores and seeks different solutions, enabling it to escape suboptimal local minima. As illustrated in Fig.1, we present two tasks, each posing distinct local minima challenges.

In *AntMaze-v1*, an ant must circumvent a central obstacle to reach its goal, with two possible routes: over the top or beneath the bottom. The optimal path depends on the ant's randomized starting position since there is only one shortest path. As shown in Fig. 11, DDiffPG discovers both routes and learns to select the shortest one based on the starting location, while baselines struggle to adjust their paths adaptively. In *AntMaze-v2*, the ant faces two goals: the top-left goal offers a higher reward, while the right-hand goal is easier to reach. Baseline models often get trapped going for the easier, lower-reward goal. However, we plot the Q-value density maps in Fig. 13, and we find that DDiffPG locates both goals and learns to go either one of them, effectively overcoming the local minima and reaching a higher cumulative return. On the other hand, RPG also explores the top-left goal and overcomes the local minima. However, it cannot consistently solve the task.

## 5.4 Ablation Studies

We investigate the impact of the number of diffusion steps, batch size, action gradient learning rate, and number of Updates-To-Data (UTD) ratio. These hyperparameters are of particular interest given the diffusion policy and our learning procedure. For batch sizes in Fig. 7(a), we observe that larger batches enhance sample efficiency. Due to the mixed multimodal batch, a larger batch-size helps smooth the learning. In Fig. 7(b), we find that the number of diffusion steps appears to have minimal impact on performance, with 5 steps being sufficient to acquire and maintain multimodal behaviors, aligning with the findings of other diffusion policy learning methods [73, 17].

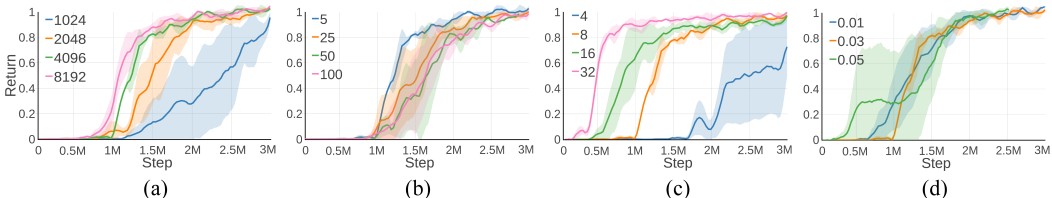

Figure 7: Abaltion studies for key parameters of DDiffPG: (a) batch size, (b) diffusion steps, (c) update-to-data (UTD) ratio, and (d) action gradient learning rate.

In Fig. 7(c), the UTD ratio indicates that more frequent updates can accelerate learning; however, a very large number would lead to increased computational demands and wall-clock time. In terms of the action-gradient learning rate (Fig.7(d)), while a higher rate initially aids learning, it may introduce increased variance due to the larger step sizes in updating target actions. An adaptive approach to the learning rate might be beneficial. Finally, our computational time analysis in Fig.8 reveals that our DDiffPG method is approximately five times slower than both TD3 and SAC, primarily due to the overhead associated with updating target actions. However, we trust that continuous developments on diffusion models will allow much faster training and inference in the future [18].

## 5.5 Online Replanning with a Multimodal Policy

We present a practical application of a trained multimodal policy in online replanning, particularly in nonstationary environments. We replicate the routine path problem described in Sec. 1 in our maze experiments by introducing random obstacles that obstruct certain paths. As discussed in Sec. 4.3, our policy can selectively execute different modes by conditioning the mode's embedding. To capitalize on this feature, we developed a proof-of-concept planner, which iteratively tests different modes until a successful route is found. Specifically, if the ant remains stationary at a location for 10 consecutive steps, the planner initiates an alternative mode. As shown in Tab. 3, while baseline methods tend to fail, becoming stuck in front of newly introduced obstacles until the end of an episode, our proof-of-concept planner demonstrates a significantly higher success rate. This demonstrates the adaptability and efficacy of our multimodal policy in performing in nonstationary environments.

## 6 Conclusions, Limitations & Future Work

In this work, we presented a novel algorithm, DDiffPG, for learning multimodal behaviors from scratch. We parameterized the policy as diffusion model and proposed diffusion policy gradient to enable training diffusion models with RL objectives in online settings. Unlike existing methods that rely on latent conditioning to retain multimodality, we emphasized explicitly discovering, preserving, and improving multimodal behaviors. First, we employed a novelty-based intrinsic motivation to explore different modes and used an unsupervised hierarchical clustering approach over trajectories to identify them. Nevertheless, the RL objective can skew the policy toward a single mode. We addressed the issue by introducing mode-specific Q-functions to optimize each mode, providing the diffusion policy with a multimodal batch to train on. We further achieved explicit mode control by conditioning the policy on a mode-specific latent embedding, which was shown to be useful for online replanning. Our evaluation demonstrated the algorithm's effectiveness in learning multimodal behaviors in complex control scenarios, such as AntMazes and robotic tasks, and showcased the potential of the multimodal policy to encourage exploration and overcome local minima.

**Limitations** However, our approach also has some limitations. *Clustering knowledge prerequisite*: Hierarchical clustering depends on a distance matrix calculated from trajectories. In practice, we compute it over the ant's 2D positions or the robot's end-effector (EE) 3D positions, rather than the entire state space. This poses a problem in scenarios like locomotion control, where different gaits may exhibit similar position changes. A possible solution is to encode trajectories into latent embeddings for clustering. *Exploration challenges*: while our current intrinsic motivation approach is effective, its performance might drop in larger spaces, as our method demands uncovering as many as possible viable solutions. This can be mitigated through different information-based motivation, or by adding a bound on the information contained withing explored modes. *Increased computation time*: the diffusion process in diffusion models lengthens training and inference wall-clock times compared to simpler MLP models, limiting, for now, the real-time application of diffusion policies in domains such as robotic control.

**Future directions** Overall, we believe our work paves the way for training multimodal diffusion policies and offers several promising research avenues. *Online replanning with long-horizon planners*: while our paper demonstrates online replanning using a multimodal policy with a brute-force method, exploring planners that can efficiently orchestrate these modes for complex, long-horizon tasks presents an intriguing opportunity. *Offline-to-online learning*: diffusion policy has been widely used in offline settings. However, the offline dataset may contain suboptimal trajectories. Our method is suitable for offline-to-online fine-tuning, allowing the agent to further refine policies without killing the previously learned modes. *Open-ended learning on large-scale environments*: viewing our method as a form of unsupervised skill discovery opens up possibilities for enabling the acquisition of diverse and useful skills beyond mere goal achievement.

# 7   Acknowledgements

This research work has received funding from the German Research Foundation (DFG) Emmy Noether Programme (CH 2676/1-1), the Daimler-Benz Foundation, and was co-financed by the EU's Horizon Europe project ARISE (Grant no.: 101135959).

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

# A    Preliminaries

**Diffusion Model**    Diffusion models [30, 67, 68] is a class of generative models that deploys a stochastic denoising process for learning to generate samples from a probability distribution $p(x)$ by mapping Gaussian noise to the target distribution through an iterative process, assuming $p_\theta(\mathbf{x}_0) := \int p_\theta(\mathbf{x}_{0:T})d\mathbf{x}_{1:T}$, where $\mathbf{x}_0, \ldots, \mathbf{x}_T$ are latent variables of the same dimensionality as the data $\mathbf{x}_0 \sim p(\mathbf{x}_0)$. A diffusion model approximates the posterior $q(\mathbf{x}_{1:T}|\mathbf{x}_0)$ through a *forward diffusion process*, i.e., a fixed Markov chain, which adds gradually Gaussian noise to the data $\mathbf{x}_0 \sim q(\mathbf{x}_0)$ according to a variance schedule $\beta_1, \ldots, \beta_T$, defined as $q(\mathbf{x}_{1:T}|\mathbf{x}_0) := \prod_{t=1}^{T} q(\mathbf{x_t}|\mathbf{x_{t-1}})$, with $q(\mathbf{x}_t|\mathbf{x}_{t-1}) := \mathcal{N}(\mathbf{x}_t; \sqrt{1-\beta_t}\mathbf{x}_{t-1}, \beta_t\mathbf{I})$. Diffusion models learn to sample from the target distribution $p(\mathbf{x}_T)$ by sampling noise from a Gaussian $p(\mathbf{x}_T) \sim \mathbf{0}, \mathbf{I}$ and iteratively denoising the noise to generate in-distribution samples, through a *reverse diffusion process* $p_\theta(\mathbf{x}_{t-1}|\mathbf{x}_t)$, defined as $p_\theta(\mathbf{x}_{0:T}) := p(\mathbf{x}_T) \prod_{t=1}^{T} p_\theta(\mathbf{x}_{t-1}|\mathbf{x}_t)$, with $p_\theta(\mathbf{x}_{t-1}|\mathbf{x}_t) := \mathcal{N}(\mathbf{x}_{t-1}; \mu_\theta(\mathbf{x}_t, t), \boldsymbol{\Sigma}_\theta(\mathbf{x}_t, t))$. The reverse diffusion process is optimized by minimizing a surrogate loss-function [30] $\mathcal{L}(\theta) = \mathbb{E}_{t\sim[1,T], \mathbf{x}_0\sim q(\mathbf{x}_0), \epsilon\sim\mathcal{N}(\mathbf{0},\mathbf{I})}\|\epsilon - \epsilon_\theta(\mathbf{x}_t, t)\|$. After training, we sample the diffusion model by $\mathbf{x}_T \sim p(\mathbf{x}_T)$ and run the reversed diffusion chain to go from $t = T$ to $t = 0$.

**Markov Decision Process**    A Markov Decision Process (MDP) is the tuple $\mathcal{M} = \langle \mathcal{S}, \mathcal{A}, \mathcal{R}, \mathcal{P}, \gamma \rangle$ [56], where $\mathcal{S}$ is the state space, $\mathcal{A}$ is the action space, $\mathcal{R} : \mathcal{S} \times \mathcal{A} \times \mathcal{S} \to \mathbb{R}$ is the reward function, $\mathcal{P} : \mathcal{S} \times \mathcal{A} \to \mathcal{S}$ is the transition kernel, and $\gamma \in [0, 1)$ is the discount factor. We define a policy $\pi \in \Pi : \mathcal{S} \times \mathcal{A} \to \mathbb{R}$ as the probability distribution of the event of executing an action $a$ in a state $s$. A policy $\pi$ induces a value function corresponding to the expected cumulative discounted reward collected by the agent when executing action $a$ in state $s$, and following policy $\pi$ thereafter: $Q^\pi(s, a) \triangleq \mathbb{E}\left[\sum_{k=0}^{\infty} \gamma^k r_{i+k+1}|s_i = s, a_i = a, \pi\right]$, where $r_{i+1}$ is the reward obtained after the $i$-th transition. Solving an MDP means finding the optimal policy $\pi^*$, i.e., the one maximizing the expected discounted return. In this paper, we are particularly interested in tasks where there are multiple goals or there is only one goal but with multiple solutions to it.

# B    Pseudoalgorithm

---

**Algorithm 1** Deep Diffusion Policy Gradient

---

1: **Input**: initial policy parameters $\theta$, initial Q-function parameters $\phi$, replay buffer $\mathcal{D}$, diffusion iteration $N$
2: **for** each iteration **do**
3:     **for** $t = 1, \cdots, T$ **do**
4:         Observe state $s_t$ and sample action $a_t^0 \sim \pi_\theta(a_t|s_t)$ via reverse diffusion process
5:         Execute action $a_t = a_t^0 + \epsilon$, where $\epsilon \sim \mathcal{N}$
6:         Initialize $a_t^{target} = a_t$
7:         Store $(s_t, a_t, a_t^{target}, r_t, s_{t+1})$ in $\mathcal{D}$
8:     **end for**
9:     **for** $g = 1, \cdots, G$ **do**
10:         Sample $M$ batch $B_i = \{(s, a, a^{target}, r, s')\}$ from replay buffer $\mathcal{D}$, where $M$ is the number of modes discovered
11:         Get $Q_{\phi_i}$, $i = 1, 2, ..., M$ from Algo. 2
12:         **for** each mode **do**
13:             Update $Q_{\phi_i}$ with Bellman Equation on $B_i$
14:             **for** $k = 1, \cdots, K$ **do**
15:                 Compute target action $a^{target}$ by one step of gradient ascent following
                    $a^{target} \leftarrow a^{target} + \eta\nabla_a Q_{\phi_i}(s, a^{target})$
16:             **end for**
17:             Replace $a^{target}$ in $\mathcal{D}$
18:         **end for**
19:         Concatenate $\{(s, a^{target})_i\}|_{i=1}^{M}$ and update diffusion policy $\pi_\theta$ following (1)
20:     **end for**
21: **end for**

---

**Algorithm 2** Mode Discovery via Clustering
---
1: **Input**: goal-reached trajectories $\{\tau^s\}$, unsuccessful trajectories $\{\tau^u\}$, previous cluster $C_{old}$
2: Compute distance matrix $D$ of $\{\tau^s\}$ with DTW metric [50]
3: Obtain cluster $C$ via hierarchical clustering with matrix $D$
4: **for** $c$ in $C$ **do**
5:     Find the cluster $c_{old}$ with the largest overlap between $c$ and $C_{old}$
6:     Assign $Q_\phi$ and $a^{target}$ from $c_{old}$ to $c$
7: **end for**
8: **for** $\tau^u$ in $\{\tau^u\}$ **do**
9:     **for** $c$ in $C$ **do**
10:        Sample $N$ trajectories from cluster $c$
11:        Compute average distance between $\tau^u$ and $\{\tau_n^s\}|_{n=1}^N$
12:     **end for**
13:     Find the cluster $c$ with the smallest average distance
14:     Add $\tau^u$ to $c$
15: **end for**
---

## C   Implementation details

Our approach includes a high-performance implementation of the proposed algorithm. First, each trajectory is assigned a unique identifier (ID), where clusters group many such IDs representing distinct modes. All trajectory elements, including states, actions, target actions, and rewards, are tagged with this ID. This allows storing all trajectories in one replay buffer, enabling efficient batch sampling. Second, computing distances between trajectories can be computationally demanding. To address this, we implement a hashmap for storing these distances, keyed by the trajectory IDs. This strategy ensures that distance computation between any two specific trajectories is performed only once.

## D   Environment details

The *AntMaze* environments are implemented based on the D4RL benchmark [20]. The *AntMaze* is a navigation task, in which the agent controls the movement of a complex 8-DOF "Ant" quadruped robot. The objective is to reach goal positions represented by the red ball(s). The robotic manipulation environments with Franka are based on [22]. The agent controls a 7-DOF Franka arm in joint space for different manipulation tasks.

For the above environments, we designed to contain multiple possible solutions to show the multimodality, either with multiple goals or multiple ways that solve the task. Note that the goal position is static and invisible to the agent, meaning that the agent has to explore the goal first. We use a **sparse reward 0-1 reward for all environments**, which is activated upon reaching the goal. We describe each environment and its multimodal solutions as follows:

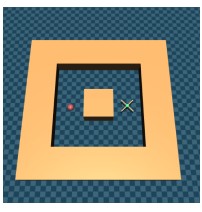

*AntMaze-v1*: it contains one goal in the maze. The ant is expected to bypass a central obstacle to reach the goal position, with two possible routes, either over the top or beneath the bottom of the obstacle. The optimal path varies depending on the ant's randomized starting position, as there is only one shortest path. The episode length is $500$.

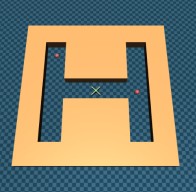

*AntMaze-v2*: it contains two goals in the maze, with the top-left goal offering a higher reward than the right-hand goal. Due to the higher reward, the optimal path is to reach the top-left goal, however, the ant may get trapped in the right goal as it is much easier to explore. The episode length is $500$.

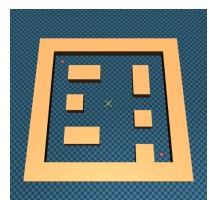

*AntMaze-v3*: it contains two goals in the maze, each accessible via multiple routes. For the **goal in the top-left**, three routes are viable: (1) go upwards to the end and then turn left; (2) move diagonally towards the top-left until encountering the left border, then head upwards, and (3) move towards the bottom-left, circumvent the left obstacle, and then go upwards. The distances of the first two paths are comparable, whereas the third is much longer. For the **goal in the right-bottom**, there are two comparable routes: (1) moving right and then downwards or (2) going downwards and then right. Possible solutions are shown in the visualization of clusetering performance in Fig. 3. The episode length is 700.

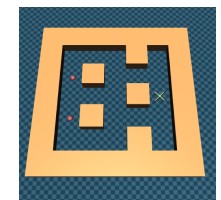

*AntMaze-v4*: it contains two goals in the maze, each accessible through two routes. For the top goal, the agent can bypass the obstacle by going up, then has the option to either go up or down to reach it. The routes to the bottom goal is symmetrical. Possible solutions are shown in Fig. 3. The episode length is 700.

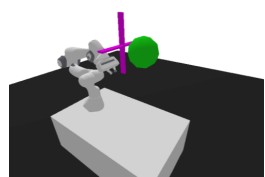

*Reach*: the agent controls the Franka arm to reach the red ball, navigating around a fixed cross-shaped obstacle that lies in the path. Despite the presence of a single goal position, the agent can bypass the obstacle in four distinct ways, offering multiple solutions to the task. The episode length is 100.

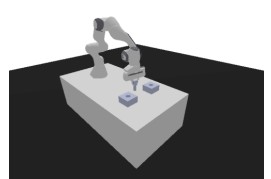

*Peg-in-hole*: the agent controls the Franka arm to perform a peg-insertion task. With two holes available on the desk, the agent can successfully complete the task by inserting the peg into either hole. The episode length is 100.

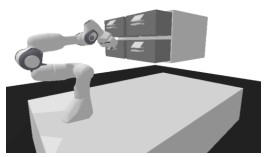

*Drawer-close*: the agent controls the Franka arm to close drawers. There are four drawers on the desk, and the agent can close either drawer to finish the task. The episode length is 100.

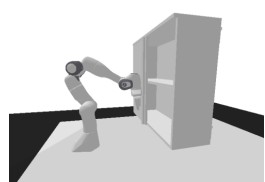

*Cabinet-open*: the agent controls the Franka arm to open a cabinet. The cabinet has two layers therefore the agent can move the arm to either layer and subsequently pull the door open to finish the task. The episode length is 100.

# E   Hyperparameters

Here, we list the hyperparameters used for all baselines and tasks.

Table 1: Hyperparameter setup for all tasks. For RPG, we use the default hyperparameters for sparse reward.

| Hyperparameter | DDiffPG/DIPO/Diffusion-QL/Consistency-AC/TD3/SAC |
|---|---|
| Num. Environments | 256 |
| Critic Learning Rate | $5 \times 10^{-4}$ |
| Actor Learning Rate | $3 \times 10^{-4}$ |
| Action Learning Rate | $3 \times 10^{-2}$ (DDiffPG/DIPO) |
| Alpha ($\alpha$) Learning Rate | $5 \times 10^{-3}$ (SAC) |
| V_min (distributional RL) | 0 |
| V_max (distributional RL) | 5 |
| Num. Atoms (distributional RL) | 51 |
| Optimizer | Adam |
| Target Update Rate ($\tau$) | $5 \times 10^{-2}$ |
| Batch Size | 4,096 |
| UTD ratio | 8 |
| Discount Factor ($\gamma$) | 0.99 |
| Gradient Clipping | 1.0 |
| Replay Buffer Size | $2,000$ trajectories $\approx 1 \times 10^6$ (DDiffPG) |
| | $1 \times 10^6$ (baselines) |
| Reclustering Frequency | 100 (DDiffPG) |
| Mode Embedding Dim. | 5 (DDiffPG) |

Table 2: Clustering threshold. The default threshold is set to $0.7 \max(Z[:, : 2])$ corresponding with MATLAB(TM) behavior, where $Z$ is the linkage matrix.

| | Clustering threshold |
|---|---|
| *AntMaze-v1* | 50 |
| *AntMaze-v2* | 70 |
| *AntMaze-v3* | 70 |
| *AntMaze-v4* | 50 |
| *Reach* | default |
| *Peg-in-hole* | default |
| *Drawer-close* | default |
| *Cabinet-open* | default |

# F Additional experimental results.

Table 3: Number of modes discovered, success rate (S.R.), and episode length (E.L.) for AntMazes and the maze with randomly initialized obstacles, averaged over 20 random seeds per case.

|  |  | DDiffPG | RPG | TD3 | SAC | DIPO | Diff-QL | Con-AC |
|---|---|---|---|---|---|---|---|---|
| *AntMaze-v1* | #modes | 2 | 1 | 1 | 1 | 1 | 1 | 0 |
|  | S.R. | 1.0 | 0.2 | 1.0 | 0.98 | 0.98 | 0.63 | 0.0 |
|  | E.L. | 75.7 | 450.1 | 59.2 | 89.1 | 75.3 | 241.7 | 500 |
| *AntMaze-v2* | #modes | 2 | 1.5 | 1 | 1 | 1 | 1 | 0 |
|  | S.R. | 1.0 | 0.5 | 0.75 | 0.53 | 0.75 | 0.59 | 0.0 |
|  | E.L. | 66.8 | 259.4 | 35.6 | 222.3 | 34.6 | 225.7 | 500 |
| *AntMaze-v3* | #modes | 4.3 | 1 | 1 | 1 | 1 | 1 | 1 |
|  | S.R. | 0.98 | 0.1 | 1.0 | 0.8 | 1.0 | 0.77 | 0.25 |
|  | E.L. | 142.5 | 642.1 | 83.4 | 207.8 | 99.3 | 226.6 | 545.1 |
| *AntMaze-v4* | #modes | 3.8 | 0 | 1 | 1 | 1 | 1 | 0 |
|  | S.R. | 1.0 | 0.0 | 1.0 | 1.0 | 1.0 | 0.75 | 0.0 |
|  | E.L. | 151.1 | 700 | 88.1 | 92.8 | 86.9 | 249.5 | 700 |
| Randomized | #modes | N.A. | N.A. | N.A. | N.A. | N.A. | N.A. | N.A. |
|  | S.R. | 1.0 | 0.0 | 0.5 | 0.45 | 0.5 | 0.45 | 0.1 |
|  | E.L. | 162.3 | 700 | 310.2 | 342.1 | 293.5 | 421.4 | 582.3 |

Table 4: Number of modes, success rate (S.R.), and episode length (E.L.) for robotic tasks, averaged over 20 random seeds.

|  |  | DDiffPG | TD3 | SAC | DIPO | Diff-QL | Con-AC |
|---|---|---|---|---|---|---|---|
| *Reach* | #modes | 2.8 | 1 | 1 | 1 | 1 | 1 |
|  | S.R. | 1.0 | 1.0 | 0.95 | 1.0 | 0.75 | 0.5 |
|  | E.L. | 23.8 | 18.0 | 20.5 | 18.6 | 40.5 | 60.5 |
| *Peg-in-hole* | #modes | 2 | 1 | 1 | 1 | 1 | 1 |
|  | S.R. | 1.0 | 1.0 | 1.0 | 1.0 | 1.0 | 0.2 |
|  | E.L. | 5.9 | 4.7 | 4.7 | 5.1 | 4.92 | 80.91 |
| *Drawer-close* | #modes | 3.5 | 1 | 1 | 1 | 1 | 1 |
|  | S.R. | 1.0 | 1.0 | 1.0 | 1.0 | 0.8 | 0.2 |
|  | E.L. | 23.6 | 22.0 | 24.7 | 22.8 | 34.5 | 80.74 |
| *Cabinet-open* | #modes | 2 | 1 | 1 | 1 | 1 | 1 |
|  | S.R. | 1.0 | 0.98 | 1.0 | 1.0 | 0.75 | 0.5 |
|  | E.L. | 21.1 | 14.3 | 24.3 | 19.6 | 42.1 | 59.5 |

We provide an evaluation of computational time compared with baselines. We use NVIDIA GeForce RTX 4090 for all experiments. However, given the intense research landscape in diffusion models, we hope to make the training and inference more time-efficient in our future work.

- For data collection, DDiffPG needs more wall-clock time than others, which is due to the trajectory processing, clustering, etc. We note that DIPO has a similar wall-clock time with the time of TD3 and SAC, implying that the impact of inference speed of diffusion model is not significant.

- For policy updates, DDiffPG and DDiffPG (v) require less computational time. This is because TD3 and SAC need to estimate the Q-value during policy updates, while DDiffPG and DIPO only need to minimize the MSE loss.

- For critic update, DDiffPG requires more wall-clock time due to multiple Q-functions.

- For target action update, only DDiffPG and DIPO needs to compute the target action. DDiffPG requires more wall-clock time because it has multimodal batches and needs to compute the target action for each sub-batch.

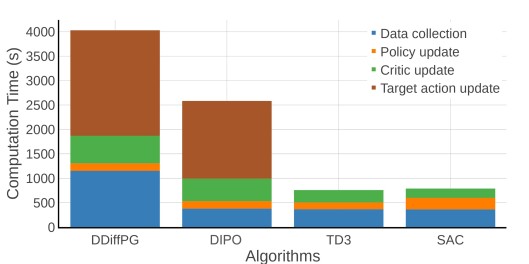

Figure 8: Comparison on wall-clock time.

We implement an alternative clustering approach on the high-dimensional state space by training a vector-quantized variational autoencoder (VQ-VAE) to learn trajectory representations. We then performed clustering using the codevectors. The visualized cluster performance and projected embedding space verify that VQ-VAE learns meaningful representations and can be used in our approach, as an alternative clustering approach. We would like to point that any unsupervised clustering method can be coupled with our approach.

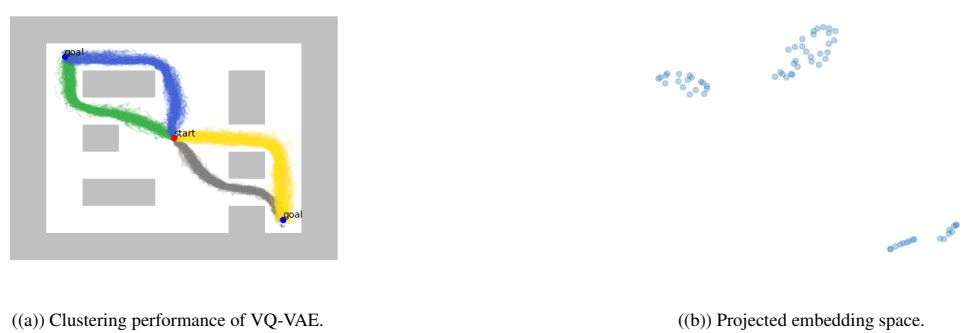

((a)) Clustering performance of VQ-VAE.                    ((b)) Projected embedding space.

Figure 9: Comparison of VQ-VAE clustering and projected embedding space.

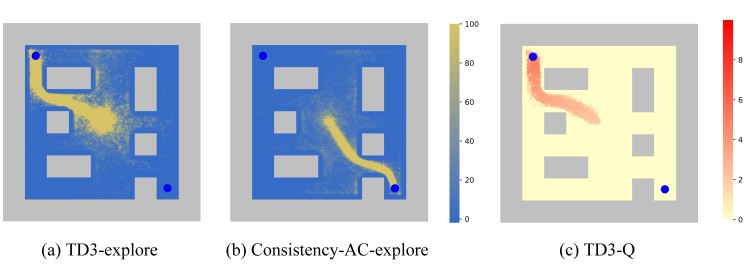

(a) TD3-explore        (b) Consistency-AC-explore        (c) TD3-Q

Figure 10: The rest of exploration maps and density maps in *Antmaze-v3*.

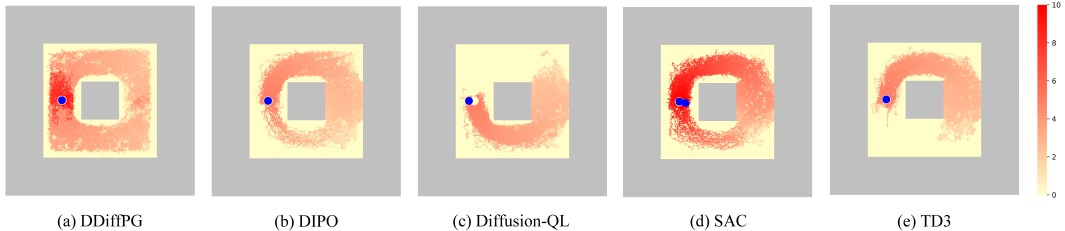

Figure 11: (a)-(d) Q-value maps of baselines and DDiffPG in *Antmaze-v1*.

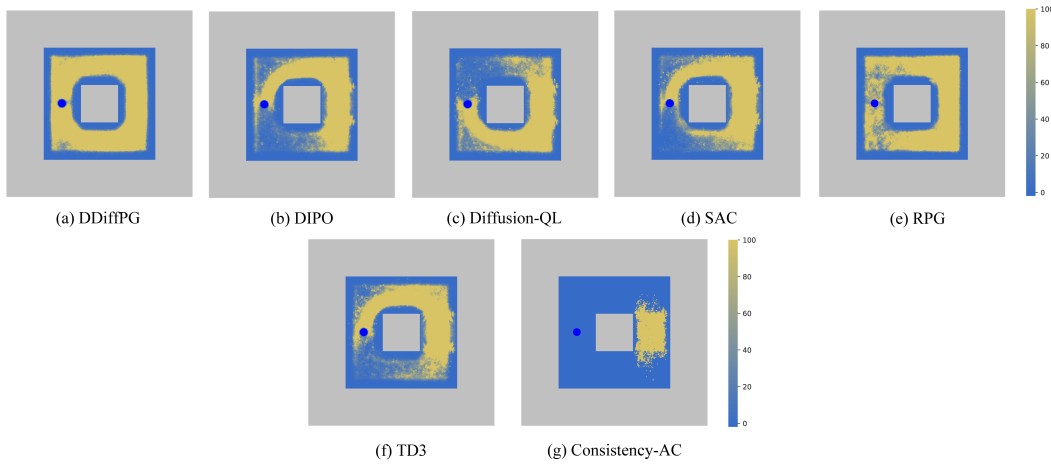

Figure 12: Exploration maps of DDiffPG and baselines in *AntMaze-v1*.

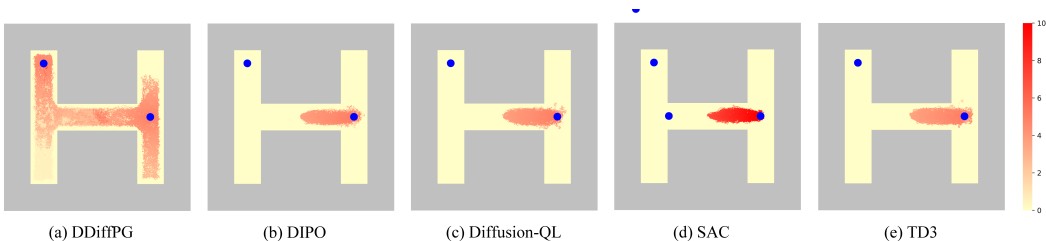

Figure 13: (a)-(d) Q-value maps of baselines and DDiffPG in *Antmaze-v2*.

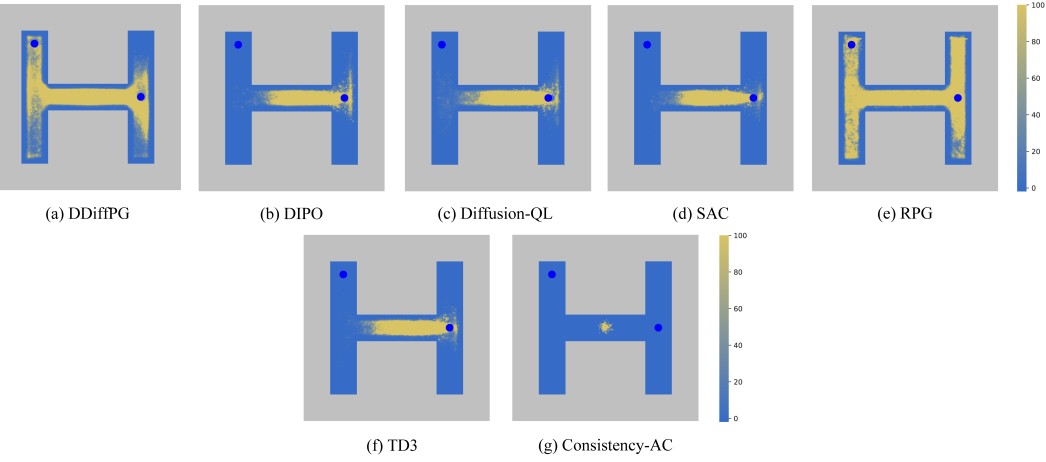

Figure 14: Exploration maps of DDiffPG and baselines in *AntMaze-v2*.

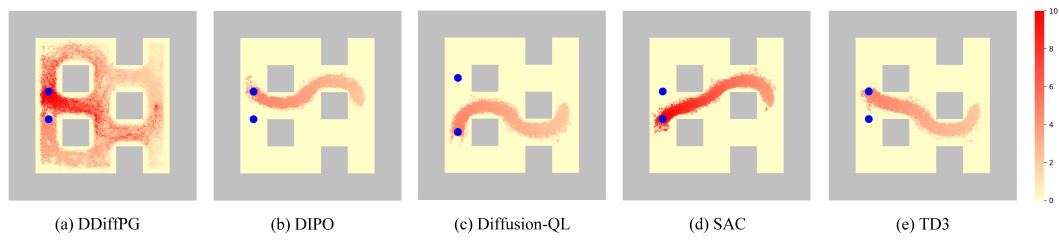

Figure 15: (a)-(d) Q-value maps of baselines and DDiffPG in *Antmaze-v4*.

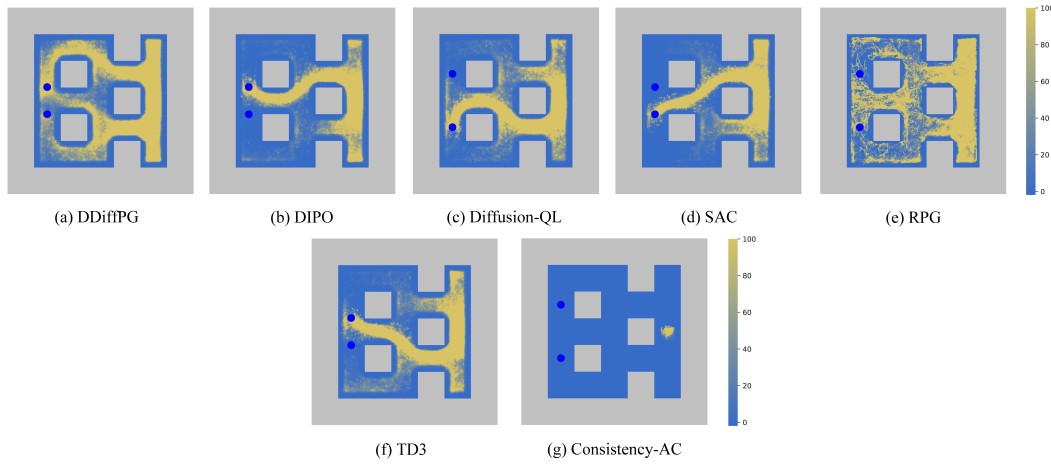

Figure 16: Exploration maps of DDiffPG and baselines in *AntMaze-v4*.

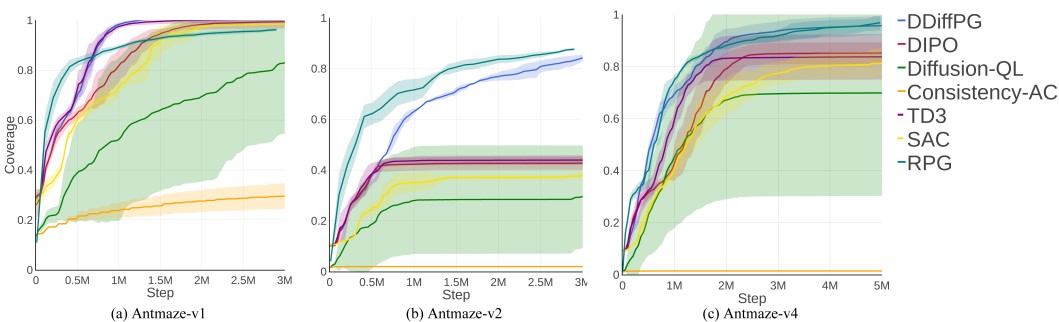

Figure 17: State coverage in *Antmaze-v1*, *Antmaze-v2*, and *Antmaze-v4*

