# OpenReview forum: "Learning Multimodal Behaviors from Scratch with Diffusion Policy Gradient"
_NeurIPS.cc/2024/Conference — NeurIPS 2024 poster_

### Official Review · Reviewer_TVD7 · 2024-06-19

**Soundness:** 3
**Presentation:** 2
**Contribution:** 3
**Rating:** 6
**Confidence:** 4

**Summary:**

This paper introduces DDiffPG for online reinforcement learning with multi-modal behaviour discovery. DDiffPG consists of two parts: 1) a new policy improvement method to stabilise the diffusion policy by cloning a target action; 2) a mode discovery mechanism to train mode-specific and intrinsic Q functions. In their experiments, the authors have shown that DDiffPG can achieve comparable performance with the baselines while producing multi-modal behaviours, which provides a series of benefits like avoiding mode collapse.

**Strengths:**

The paper has introduced an interesting idea. To the best of my knowledge, this is the first work that allows diffusion policy to learn multi-modal behaviours during online RL. According to the experiments, the proposed method has produced a reasonable performance with nice multi-modal behaviours. Besides, the paper has provided nice visualisations and discussions to help understand the proposed approach.

**Weaknesses:**

There are several main weaknesses of the paper.

- The paper is hard to follow and the presentation has a certain room for improvement.

- In section 3, the formal theoretical derivation of the newly introduced policy improvement objective is missing. Although it shows that this method worked empirically, it remains unclear how the resulting policy theoretically maximises the expected return in general.

- I feel the paper is a bit over-claiming for certain aspects. In section 5.3, the authors claimed that DDiffPG can *overcome* local minimum issues and encourage exploration. However, the exploration comes from the use of RND when learning $Q_\mathrm{explore}$, rather than the architecture itself. In addition, it is a very strong claim that DDiffPG **overcomes** the local minimum issues. The experiments are conducted on only 8 state-based tasks, from my point of view, which is insufficient to support such a general claim. I understand that by capturing multi-modal distributions, DDiffPG allows better generalisation, but I would suggest the authors moderate the claims a bit.

Minor issues:
- In line 157, is this a typo? In $r^\mathrm{intr}(s, a, s’) = \max(\mathrm{novelty}(s’) - \alpha \mathrm{novelty}(s’), 0)$, should this be $r^\mathrm{intr}(s, a, s’) = \max(\mathrm{novelty}(s’) - \alpha \mathrm{novelty}(s), 0)$?

**Questions:**

- In section 4.2, line 183, I’m not fully convinced that the RL objective can skew the policy towards a single mode. Suppose we have a Q function that nicely captures two modes. During policy improvement, let’s say we sampled a batch of trajectories that equally captures both modes, and we perform policy improvement by $\max \mathbb{E}\left[Q(s, a)\right]$. Given that our Q function already nicely captures both modes, why does such an objective cause mode collapse? Could you provide more explanations? Considering the success of DDiffPG on capturing the multi-modality in the policy space, is this really because of the way you perform policy improvement in Eqn. 1, or is it because the DDiffPG used multiple Q functions for separate modes, which just better fits the multi-modal distribution?

- Regarding the use of mode-specific Q functions, it is a bit unclear to me how to stabilise the training. One issue is that during online exploration, the dataset is continuously being updated and modes are being updated. In this case, how do we fix the correspondence between the Q functions being learned and the mode? Besides, according to line 167, DDiffPG requires no predefined number of clusters, and the number of modes could be dynamic. However, we have to initialise a fixed number of Q functions. This seems a bit contradictory to me. How to define the number of Q functions during training?

- It seems to me the exploration is only guaranteed by the training of $Q_\mathrm{explore}$ using RND. However, how do we balance the exploration and exploitation during RL?

**Limitations:**

- In line 175, the definition of DTW requires task privileged information, e.g., object positions. This is a major issue and unrealistic. In more realistic scenarios, people normally have no access to such information, and as a result, this clustering approach is not applicable.
- The paper only conducted experiments on 8 simple state-based environments. It is unclear how the approach could generalise to more realistic environments and tasks with visual observations. Although at the current stage, I understand that the current results reasonably demonstrate the capability of the proposed method, more realistic tasks will better support many claims made by the paper.

---

> ### Author Rebuttal · Authors · 2024-08-07
>
> We thank the reviewer for their feedback. Due to the rebuttal limit and the number of questions, our responses are concise. We are happy to provide more detailed answers during the discussion period.
>
> > The paper is hard to follow ... for improvement.
>
> Based on the reviewer's feedback, we will:
> 1. Moderate our claim about overcoming local minima and clarify DDiffPG's continuous exploration ability in Sec. 5.3.
> 2. Fix the typo in the intrinsic reward definition.
> 3. State clearly scenarios where an imperfect Q-function can lead to mode collapse.
> 4. Highlight the reclustering process in Sec. 4.2 and the exploration-exploitation balance in Sec. 4.3.
> 5. Clarify the information used in DTW computation, such as robot positions.
>
> > In section 3, ... in general.
>
> In our method and concurrent works (Yang et al. 2023, Psenka et al. 2024), policy update for a diffusion process cannot follow the usual deterministic policy gradient derivation. The stochastic nature of diffusion models poses significant challenges in deriving formal theoretical guarantees of policy improvement. However, our approach updates target actions under the guidance of the Q-function. As the Q-function improves, these target actions are refined accordingly, which in turn trains the diffusion policy on these improved targets. While formal proof remains challenging, this intuitive process reflects an alignment between the policy and the evolving Q-function, suggesting effective policy improvement in practice.
>
> > I feel the paper ... moderate the claims a bit.
>
> We apologize for any ambiguity in our claim. **We mean that the continuous exploration capability of DDiffPG helps increase state coverage and overcome local minima**. This capability comes from the exploratory mode, not from RND alone, as the **same intrinsic reward** was used **for all baselines**. DDiffPG maintains an exploratory mode throughout training, allowing it to discover versatile behaviors continuously. Unlike baselines that quickly converge to the first explored solution, DDiffPG keeps exploring, which is crucial for learning multimodal behaviors online. Fig. 6 and 12 support our claim, showing that DDiffPG achieves the highest state coverage, while overcoming the lower-reward goal, reaching the top-left goal with the higher reward, helping to escape the local minimum.
>
> > In line 157, is this a typo? ...
>
> Yes, the intrinsic reward is defined as the novelty difference between $s'$ and $s$, which encourages to visit the boundary of explored region.
>
> > In section 4.2, ... distribution?
>
> We agree that a perfect Q-function nicely capturing both modes could prevent mode collapse. However, in practice, achieving a perfect Q-function is unlikely since both the policy and Q-function are learned from scratch. This often leads to:
>
> 1. the policy being guided towards the first explored goal.
> 2. the policy converging to the solution with higher Q-values due to the discount factor, even if two modes are initially captured.
>
> DDiffPG's design mitigates mode collapse through mode-specific Q-functions, each focused on a single mode, thereby isolating their Q-values. By sampling actions from the replay buffer rather than relying solely on policy-inferred actions, we ensure balanced improvement across all modes. However, **policy improvement alone is not sufficient for multimodal behavior learning**, as confirmed by the results of DIPO (Tab. 3 and 4).
>
> > Regarding the use of mode-specific ... during training?
>
> Yes, we re-cluster trajectories every $F$ iterations, updating modes continuously. After re-clustering, new clusters inherit Q-functions and $a^\text{target}$ from the most overlapping cluster in the previous iteration. If the number of clusters increases, we initialize new Q-functions. This ensures a one-to-one correspondence between modes and Q-functions without needing to predefine a fixed number of Q-functions. This process is explained in Sec. 4.2 (lines 210-214), with further details in Alg. 2 and App. C. The implementation can also be found in our provided code to the Meta-reviewer under `ddiffpg/utils/Q_scheduler.py`.
>
> > It seems ... during RL?
>
> We balance exploration-exploitation by adjusting the proportion of the exploratory mode during data collection. As explained in Sec. 4.3 (lines 230-233), we condition the diffusion policy on mode-specific embeddings, including an exploratory mode trained by $Q_\text{explore}$. We set the proportion for exploration to $p = \max(0.3, 1/\text{modes})$, ensuring at least 30% of actions are exploratory. Implementation details can be found in `ddiffpg/algo/ddiffpg.py` (lines 104-129).
>
> > In line 175, ... not applicable.
>
> We apologise for any confusion. **We only use the robot position for clustering, i.e., the Ant's position and the robot end-effector's position, which are accessible in real-world scenarios**. We do not use any information about objects or goals in the environment. In the PDF, we provide an alternative clustering approach by training a VQ-VAE to learn trajectory representations and using the codevectors for clustering, which learns meaningful representations suitable for our approach.
>
> > The paper only ... by the paper.
>
> We want to emphasize that the 8 robot locomotion and manipulation tasks are nontrivial. They are all **high-dimensional and continuous control** tasks with **sparse rewards**. In the AntMaze tasks, the agent must solve the Ant-locomotion task through joint control, a notoriously difficult continuous control problem, while also learning to steer locomotion gaits to find goals in complex mazes. All tasks are **goal-agnostic**. The agent must explore to find the goal, making it challenging to learn multimodal behaviors due to the vast exploration space. As the first work exploring online multimodal diffusion policies, we see our current tasks as a strong starting point and plan to extend our approach to long-horizon and vision-based tasks with greater multimodality in future work.

---

> > ### Comment · Reviewer_TVD7 · 2024-08-09
> >
> > I appreciate the detailed responses by the authors. Most of my concerns have been addressed. I will increase my rating to 6.

---

> > > ### Comment · Area_Chair_ZbvM · 2024-08-12
> > > **Reviewer Discussion Requested**
> > >
> > > Dear Reviewer,
> > >
> > > The discussion time is coming to an end soon. Please engage in the discussion process which is important to ensure a smooth and fruitful review process. Give notes on what parts of the reviewers responses that have and have not addressed your concerns and why.

---

### Official Review · Reviewer_HLNL · 2024-07-04

**Soundness:** 3
**Presentation:** 3
**Contribution:** 3
**Rating:** 7
**Confidence:** 4

**Summary:**

This paper addresses the challenges associated with employing diffusion policy in online reinforcement learning (RL), particularly the intractability of policy likelihood approximation and the bias towards a single mode. The author introduces the Deep Diffusion Policy Gradient (DDiffPG) method, which decouples exploration from exploitation. For exploration, novelty-based intrinsic motivation and hierarchical clustering are utilized to identify modes, while for exploitation, the author describes the mode-specific Q-function and a multimodal data batch. Empirical evaluations demonstrate that DDiffPG effectively masters multimodal behaviors.

**Strengths:**

+ The application of diffusion policy for multiple modes in an online setting is promising and addresses a previously unexplored area in the literature.
+ The introduction of a diffusion-based policy gradient method is novel and represents a significant contribution to the field.
+ The work is well-motivated, and the visualization of multimodal behaviors using antmaze examples effectively enhances understanding and illustrates the practical utility of the approach.

**Weaknesses:**

+ Several claims require additional support. For instance, the author asserts that standard exploration-exploitation strategies may easily converge towards a single mode (Lines 25-27) without providing theoretical or experimental evidence. Similar issues are present in Lines 35-36 and Lines 52-53. These statements are crucial for constructing the paper's motivation and thus require more substantial support to enhance their reliability.

**Questions:**

+ In Lines 263-267, the author explains that DDiffPG could learn suboptimal paths. Is this statement intended to justify the suboptimal performance compared to TD3? The author suggests that this suboptimal issue can be mitigated by the mode embeddings. It would be more effective to present the best performance and use the suboptimal trajectories as ablations, specifically when blocking the optimal path, to highlight the significance of multiple trajectories.
+ Why does directly using the action gradient to optimize the policy lead to vanishing gradients and instability? Is this due to the large denoising steps? Including corresponding ablation studies would provide a better illustration.
+ Unlike the offline setting where trajectories are stable, the replay buffer with the updated Q function results in changed pairs of $(s, a^{target})$. Does training the diffusion model with a supervised framework on continually changing pairs lead to instability in learning? (Lines 126-129)
+ Why does DIPO, which uses the original $a$ from the buffer, not know the true outcome? I understand that the replay buffer contains the past trajectories $(s,a,r,s')$ (Lines 134-137).
+ Are the mode-specific Q-functions also applicable to other standard policies?

**Limitations:**

--

---

> ### Author Rebuttal · Authors · 2024-08-07
>
> We thank the reviewer for their positive feedback and insightful comments. We want to address the reviewer's concerns and questions as follows.
> > Several claims require additional support. ... enhance their reliability.
>
> We thank the reviewer for the constructive comment. First, given the RL objective of maximizing the expected return, policies in goal-oriented tasks will converge to a solution once it is discovered. Tab. 3 and 4 experimentally show that traditional RL baselines fail to learn multimodal behaviors even when multiple solutions exist.
>
> Second, DDiffPG maintains an exploratory mode throughout training, allowing it to continuously discover versatile behaviors. Fig. 6 demonstrates that DDiffPG achieves the highest state coverage, supporting our claim. Additionally, Fig. 12 illustrates a local minima scenario where the ant faces two goals. Baseline models often get trapped pursuing the easier, lower-reward goal, whereas DDiffPG continues to explore and eventually learns to reach the higher-reward goal, helping to escape the local minimum.
>
> Third, DDiffPG achieves explicit mode discovery via hierarchical clustering with the DTW metric, preserves the multimodal action distribution by constructing multimodal training batches for policy updates, and improves the modes collectively with mode-specific Q-functions.
>
> > In Lines 263-267, ... significance of multiple trajectories.
>
> We agree with the reviewer that presenting the best performance is beneficial. DDiffPG achieves the same success rate as TD3 in Fig. 4 but usually requires more steps to reach the goal in Tab. 3 and 4. In response, we provide the episode lengths that execute only the mode with the shortest path below. We will include these results in Tab. 3 and 4 in the revised version. Note that further training of DDiffPG would lead to further optimizing all paths. We report here the numbers obtained by the policies reported in the paper.
>
> ||DDiffPG|TD3|
> |-|-|-|
> |Maze-v1|65.3|59.2|
> |Maze-v2|40.9|35.6|
> |Maze-v3|79.8|83.4|
> |Maze-v4|103.5|88.1|
> |Reach|19.1|18.0|
> |Peg-insertion|5.0|4.7|
> |Drawer-close|20.5|22.0|
> |Cabinet-open|16.8|14.3|
>
>
> > Why does directly ... better illustration.
>
> We thank the reviewer for the insightful suggestions. Directly backpropagating the action gradient through the diffusion model for policy optimization is challenging and can cause instability due to the Markov chain in the diffusion process and its stochastic nature [1, 2, 3]. This issue is evident in the high variance of the baseline Diffusion-QL in Fig. 4. Additionally, we have included plots in the attached PDF to measure the gradient in Diffusion-QL, confirming the vanishing gradient issue.
>
> [1] Psenka, Michael, et al. "Learning a diffusion model policy from rewards via q-score matching." ICML, 2024.
>
> [2] Wallace, Bram, et al. "Diffusion model alignment using direct preference optimization." CVPR, 2024.
>
> [3] Pascanu, Razvan, et al. "On the difficulty of training recurrent neural networks." ICML, 2013.
>
>
> > Unlike the offline setting ... (Lines 126-129)
>
> In the proposed diffusion policy gradient, we can tune the step size $\eta$ of the action gradient for stability. In Section 5.4, we provided a hyper-parameter analysis on the step size and found that a step size of 0.03 can generally work well. However, an aggressive value, e.g., 0.05, could speedup the learning at the early stage but increases the variance. Consequently, we used a step size of 0.03 for all tasks without any further tuning.
>
> > Why does DIPO, ... (Lines 134-137).
>
> According to Algorithm 2 in [4], DIPO stores transition $(s_t, a_t, s_{t+1}, r(s_{t+1}|s_t, a_t))$ into the replay buffer, performs gradient ascent on $a_t$, and replaces $a_t$ in the original buffer. Therefore the transition $(s_t, a_t, s_{t+1}, r(s_{t+1}|s_t, a_t))$ no longer aligns with the current MDP dynamics and reward function, due to the replacement of $a_t$. Given that DIPO is an off-policy algorithm, the reuse of these replaced transitions for training the Q-function could be problematic, as the agent is training values of actions that have not been actually played out in the environment, and their true outcome (reward and next state) are unknown.
>
> [4] Yang, Long, et al. "Policy representation via diffusion probability model for reinforcement learning." arXiv, 2023.
>
> > Are the mode-specific ... standard policies?
>
> We agree with the reviewer that the mode-specific Q-functions may be applicable to other candidate model parameterizations. The most straightforward example is to learn a separate unimodal actor for each Q-function. However, we advocate for the benefits of learning a unified model:
> 1. Learning a single model allows to share information (e.g., representations) across modes, which is particularly beneficial for future research, e.g., on image-based-observation tasks [5]. Our multimodal policy learning also draws an analogy to multitask RL, in which the objective is to solve multiple tasks with a single policy rather than separate policies per task, benefiting from knowledge sharing [6].
> 2. Learning separate unimodal policies would significantly increase the computational time and agent interactions with the environment. With separate policies, we need to iteratively update them; however, only one backpropagation is needed for a single multimodal policy.
> 3. When a new mode is discovered, our diffusion model continues learning, adding this mode in its landscape, without forgetting previous knowledge. However, if we have separate policies, we have to initialize a new policy and learn from scrach, otherwise additional techniques have to be introduced to transfer knowledge from previous training.
>
> [5] Kalashnikov, Dmitry et al. “QT-Opt: Scalable Deep Reinforcement Learning for Vision-Based Robotic Manipulation.” CoRL, 2018.
>
> [6] Hendawy, Ahmed et al. “Multi-Task Reinforcement Learning with Mixture of Orthogonal Experts.” ICLR, 2024.

---

> > ### Comment · Reviewer_HLNL · 2024-08-09
> >
> > I have reviewed the authors' rebuttal and their responses to the other reviewers, which have addressed my concerns. I believe this paper has a good contribution to online diffusion policy learning, particularly in multimodal policy. Given the engineering focus, I agree that extensive proofs are not necessary. I will adjust my score accordingly.

---

### Official Review · Reviewer_U1rA · 2024-07-09

**Soundness:** 2
**Presentation:** 3
**Contribution:** 3
**Rating:** 5
**Confidence:** 4

**Summary:**

This paper aims to solve online RL problems with diffusion policy. It includes 1. a diffusion policy optimization method for diffusion online training. 2. A combination of intrinsic rewards motivated skill discovery method and model-seeking Q-learning to facilitate exploration and prevent mode-collapse behavior. 3. Several self-designed environments where there might  be multiple optimal solutions and thus require expressive exploration policy.

**Strengths:**

1. The paper shows diffusion policy has a big potential in online RL because it enables multimodal exploration.
2. The self-designed environments are a good contribution to the research field by showcasing the necessity of diffusion exploration.
3. Performance clearly surpasses several baselines.

follow up:

The experiments basically support comments in the paper.  The paper sets out to handle the single-mode exploration problem in online RL, and the self-designed environments, unlike most previous classics, allow diverse optimal behaviors and can benefit from multimodal exploration. The experiments show that the proposed method outperforms several classic baselines including some diffusion-based methods.

**Weaknesses:**

1. The proposed diffusion training objective seems handcrafted and requires a lot of tunning. This may limit the algorithms' further application.
2. Besides the diffusion optimization methods. Other proposed techniques are more like a good combination of previous work. This indicates limited theoretical novelty.
3. Code is not provided. For this style of paper, I think code quality is essential, and a mere promise to release the code is not convincing.

follow up:
1. The ablation studies are not strong enough to prove the improved performance number actually comes from multimodal exploration. I cannot be certain which part of the method works from the experiments.  More visualization/empirical results/analyses should be given.
2. The formatting of the table/figure can be greatly improved. For instance, the title of Figure 4 is wrong/incomplete. Table 3/4 is referenced as the main results in the paper but only put in the appendix.
3. The diffusion optimization results also lack very strong novelty. The loss function is basically a supervised learning loss adapted for online RL, without strong convergence or policy improvement guarantee. Still, the diffusion+online RL theories are a known unsettled and hard problem, so this kind of exploration is fine and meaningful.

**Questions:**

None

---

> ### Author Rebuttal · Authors · 2024-08-07
>
> We thank the reviewer for the positive feedback and insightful comments. We address the reviewer's concerns and questions as follows.
>
> > The proposed diffusion training objective ... further application.
>
> We proposed diffusion policy gradient, a method that combines RL and behavioral cloning (BC) for stable online diffusion policy updates. Inspired by methods like the Deterministic Policy Gradient (DPG) [1], the policy is optimized to follow the action-gradient $\nabla_{a} Q(s, a)$.
>
> Directly backpropagating the action gradient through the diffusion model for policy optimization is known to be difficult and may cause the gradient vanishing problem and instability [2, 3]. This issue was evident in the high variance of the baseline Diffusion-QL in Fig. 4. We have also included additional plots to visualize this issue in the attached PDF. To mitigate this issue, we imitate the $a^{target}$ obtained via action gradient ascent. This way, we obtain an action that follows the updated Q and serves as the target for our BC objective. The solution we provided is general and requires no tuning. This can also be seen in the codebase we provided to the Meta-reviewer.
>
> We validated our method on 8 challenging robotic tasks with high-dimensional and continuous action spaces, requiring no handcrafting of the policy update objectives. The proposed objective only introduces an additional hyper-parameter, which is the step size $\eta$ of the action gradient. In Sec. 5.4, we provided a hyper-parameter analysis on the step size and found that a step size of 0.03 can generally work well. As a result, we used this value for all tasks without any further tuning, proving the generality of the proposed method.
>
> > Besides the diffusion ... limited theoretical novelty.
>
> While we acknowledge that some concepts used in our paper have been explored in prior works, **the combination of these techniques in an online setting to learn multimodal behaviors using diffusion policy has not been studied before**. For instance, DIPO [4] and QSM [2] focused only on the diffusion policy update objective. They do not study how to take advantage of the diffusion model to represent multimodal action distributions, nor do they consider the additional exploration challenges for discovering and learning multiple behavioral modes online. Our experiments show that naively learning a diffusion policy does not yield multimodal policies and might even be an "overkill" if multimodality is not a concern. Additionally, we highlighted the issue of the greedy RL objective in learning multimodal behaviors, which was not examined in prior works.
>
> > Code is not provided. ... the code is not convincing.
>
> We agree with the reviewer on the importance of open-source code for the community. We have sent the code link of an anonymous repo to the AC, and kindly suggest reaching out to them for access to review it.
>
> > The ablation studies ... should be given.
>
> First, we would like to emphasize that our main contribution is learning diffusion policies that capture multimodal behavioral modes. Tab. 3 and 4 show that our approach uniquely masters diverse behaviors. Despite exploring a larger space, Fig. 4 demonstrates that DDiffPG achieves comparable sample efficiency to baselines across all 8 tasks.
>
> Second, we included DIPO as a baseline, which also serves as an ablation of our framework. As discussed in Sec. 3, Lines 130-137, we modified DIPO for consistency in MDP dynamics and reward function. The DIPO lacks clustering, mode-specific Q-functions, and multimodal batch. However, DIPO fails to explore and learn multimodal behaviors, proving the necessity of our design choices.
>
> Third, we appreciate the reviewer's constructive suggestions and provide additional ablations on mode-specific Qs and the multimodal batch. The attached PDF shows that a single Q-function cannot address the greedy RL objective, guiding the policy toward the first explored solution. Moreover, in the ablation of multimodal batch, we find that while the diffusion policy learns multiple solutions, the modes are imbalanced due to the lack of enforced diversity in the batch, causing minority modes to appear less frequently. Thus, both design choices are crucial for learning multimodal behaviors.
>
> > The formatting ... in the appendix.
>
> We thank the reviewer for pointing out these inconsistencies. We will update the caption of Figure 4 to "Performance of DDiffPG and baseline methods in the four AntMaze and robotic manipulation environments." Due to space limitations, we have placed Tables 3 and 4 in the appendix. Since the camera-ready version allows for one additional page, we will include these tables in the main body upon acceptance.
>
> > The diffusion ... fine and meaningful.
>
> In our methodology and concurrent works [2, 4], policy update for a diffusion process cannot follow the usual deterministic policy gradient derivation. The stochastic nature of diffusion models poses significant challenges in deriving formal theoretical guarantees of policy improvement. However, intuitively, our approach updates target actions under the guidance of the Q-function. As the Q-function improves, these target actions are refined accordingly, which in turn trains the diffusion policy on these improved targets. While a formal proof remains challenging, this intuitive process reflects an alignment between the policy and the evolving Q-function, suggesting effective policy improvement in practice.
>
> [1] Silver, David, et al. "Deterministic policy gradient algorithms." ICML, 2014.
>
> [2] Psenka, Michael, et al. "Learning a diffusion model policy from rewards via q-score matching." ICML, 2024.
>
> [3] Wallace, Bram, et al. "Diffusion model alignment using direct preference optimization." CVPR, 2024.
>
> [4] Yang, Long, et al. "Policy representation via diffusion probability model for reinforcement learning." arXiv, 2023.

---

> > ### Comment · Reviewer_U1rA · 2024-08-08
> > **Reponse**
> >
> > I thank the authors for giving me detailed responses, which have resolved some of my concerns.
> >
> > Overall, I think the paper is above the acceptance threshold, though marginally. I thus maintain my score.

---

> > > ### Comment · Area_Chair_ZbvM · 2024-08-12
> > > **Reviewer Response Needed**
> > >
> > > Dear Reviewer,
> > >
> > > The discussion time is coming to an end soon. Please engage in the discussion process which is important to ensure a smooth and fruitful review process. Give notes on what parts of the reviewers responses that have and have not addressed your concerns and why.

---

### Author Rebuttal · Authors · 2024-08-07

We would like to thank all reviewers for their feedback and constructive suggestions on our manuscript. We are glad that the reviewers find our paper to be:
* introducing a novel and interesting idea (Reviewer HLNL, Reviewer TVD7)
* having informative and good-quality visualizations and experiments (Reviewer U1rA, Reviewer HLNL, Reviewer TVD7)
* showcasing superior performance againt baselines (Reviewer U1rA, Reviewer TVD7)
* having practical impacts on the community (Reviewer U1rA, Reviewer HLNL)

In response to the reviewers concerns we add the following responses

### Code Release

As Reviewer U1rA suggested, we sent an anonymous repo to the meta-reviewer to be distributed internally.

### Additional experiments in attached PDF
1. As Reviewer U1rA suggested, we conducted additional ablations on mode-specific Q-functions and multimodal training batch. Based on the state visitation that demonstrates the behavior of the policy during training and the behaviors in evaluation, we found that a single Q-function cannot address the issue of greedy RL objective. For the ablation of the multimodal batch, while the diffusion policy learns multiple solutions, the modes are imbalanced due to the lack of enforced diversity in the batch, causing minority modes to appear less frequently. Thus, both design choices are crucial for learning multimodal behaviors.
2. As Reviewer HLNL suggested, we measure the actor gradient norm between our approach and Diffusion-QL to verify the gradient vanishing problem. As shown in the figure, the actor gradient of Diffusion-QL is very small and particularly, the high variance in the results reported in Fig. 4 indicates that the gradient of certain seeds remains zero throughout the training.
3. As Reviewer TVD7 suggested, we implement an alternative clustering approach on the high-dimensional state space by training a vector-quantized variational autoencoder (VQ-VAE) to learn trajectory representations. We then performed clustering using the codevectors. The visualized cluster performance and projected embedding space verify that VQ-VAE learns meaningful representations and can be used in our approach, as an alternative clustering approach. We would like to point that any unsupervised clustering method can be coupled with our approach.

The figures and results are included in the attached PDF. Next, we will address each reviewer’s concerns individually.

---

### Decision · Program_Chairs · 2024-09-25

**Decision:**

Accept (poster)

**Comment:**

This work proposes a novel combination of DDPG and diffusion. The work overcomes some of the challenges in training diffusion policies in online learning settings. The reviewers agree that the contribution is novel and an interesting way to create a policy that can better explore multi-modal behaviors. However, the authors should be careful with wording and overclaiming with respect to the limitations of prior methods that may or may not be able to overcome multi-modal exploration.